# LEARNING A BEHAVIORAL REPERTOIRE FROM DEMONSTRATIONS

## ABSTRACT

Imitation Learning (IL) is a machine learning approach to learn a policy from a set of demonstrations. IL can be useful to kick-start learning before applying reinforcement learning (RL) but it can also be useful on its own, e.g. to learn to imitate human players in video games. However, a major limitation of current IL approaches is that they learn only a single "average" policy based on a dataset that possibly contains demonstrations of numerous different types of behaviors. In this paper, we present a new approach called *Behavioral Repertoire Imitation Learning* (BRIL) that instead learns a *repertoire of behaviors* from a set of demonstrations by augmenting the state-action pairs with behavioral descriptions. The outcome of this approach is a single neural network policy conditioned on a behavior description that can be precisely modulated. We apply this approach to train a policy on 7,777 human demonstrations for the build-order planning task in StarCraft II. Dimensionality reduction techniques are applied to construct a low-dimensional behavioral space from the high-dimensional army unit composition of each demonstration. The results demonstrate that the learned policy can be effectively manipulated to express distinct behaviors. Additionally, by applying the UCB1 algorithm, the policy can adapt its behavior – in-between games – to reach a performance beyond that of the traditional IL baseline approach.

## 1 INTRODUCTION

Deep Reinforcement learning has shown impressive results, especially for board games (Silver et al., 2017) and video games (Mnih et al., 2015). However, reinforcement learning (RL) has critical shortcomings when reward signals are sparse or interactions with the environment are expensive. There are several attempts to mitigate these shortcomings, including curriculum learning (Bengio et al., 2009; Graves et al., 2016; Matiisen et al., 2017), reward shaping (Ng et al., 1999), curiosity-driven exploration (Pathak et al., 2017), diversification (Conti et al., 2018; Eysenbach et al., 2018), and Imitation Learning (Bakker & Kuniyoshi, 1996).

In this paper, we focus on Imitation Learning (IL), wherein the goal is to learn a policy from a dataset of demonstrations, possibly coming from a human, another artificial system, or a collection of different entities. IL can be combined with RL, either to kick-start the learning process with IL and then improving the policy further with RL (Silver et al., 2016) or by running both methods in parallel (Harmer et al., 2018). Traditional IL techniques result in a single policy, which usually expresses an "averaged" behavior among all the behaviors present in the training data. We see this as a major limitation of IL. It would be more desirable to instead learn a diverse set of policies, expressing all the different types of behaviors present in the dataset. Additionally, having a repertoire of different behaviors allows a system to adapt to changes when it is deployed.

Addressing the limitations of current IL methods, we present a new IL approach called *Behavioral Repertoire Imitation Learning* (BRIL), which is inspired by Quality-Diversity (QD) algorithms (Pugh et al., 2016; Mouret & Clune, 2015) and RL methods that learn multiple different behaviors. In contrast to traditional optimization techniques, QD-algorithms attempt to find a diverse set of high-quality solutions rather than a single optimal solution. When QD-algorithms search in policy space, they typically discover hundreds or thousands of different policies controlled by different neural networks. BRIL instead learns a behavioral repertoire using *a single model* that can be manipulated to express multiple behaviors, similarly to RL algorithms that learn a single policy for

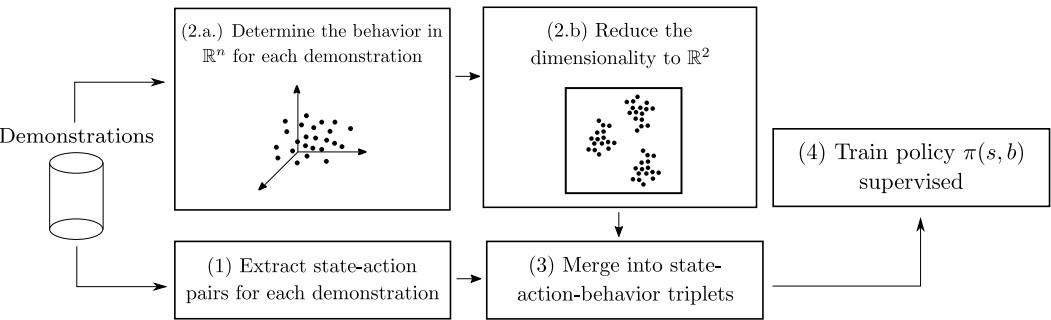

Figure 1: **Behavioral Repertoire Imitation Learning (BRIL)** trains a policy $\pi(s, b)$ supervised on a data set of state-action pairs augmented with behavioral descriptors in $\mathbb{R}^2$ for each demonstration. When deployed, a system can adapt its behavior by modulating $b$. High-dimensional behavioral spaces can be reduced using dimensionality reduction, as low-dimensional behavioral descriptions allow for faster adaption.

multiple goals (Schaul et al., 2015; Andrychowicz et al., 2017). BRIL consists of a multi-step process (see Figure 1) wherein the experimenter: (1) extracts state-action pairs (similarly to many IL approaches), (2) designs a set of behavioral dimensions to form a behavioral space (similarly to many QD algorithms) and determines the behavioral description (coordinates in the space) for each demonstration, (3) merges the data to form a dataset of state-action-behavior triplets, and (4) trains a model to predict actions from state-behavior pairs through supervised learning. When deployed, the model can act as a policy and the behavior of the model can be manipulated by changing its behavioral input features.

BRIL is tested on the build-order planning problem in StarCraft, in which a high-level policy controls the build-order decisions for a bot that has otherwise scripted modules for low-level tasks, similarly to Churchill & Buro (2011); Justesen & Risi (2017). We show that the learned policy can be optimized online by modulating the behavioral features using the Upper Confidence Bounds (UCB1) algorithm, such that it outperforms the traditional IL approach against a fixed opponent. We believe this approach can be particularly useful when modeling human players in a game by expressing the entire range of distinct behaviors instead of the average of all. We hypothesize that this property can allow a system to be more robust to exploitation, which is a concern for AI systems in many games. Furthermore, BRIL could be useful in applications beyond games, such as adaptive and resilient robotics.

## 2 BACKGROUND

### 2.1 IMITATION LEARNING

While Reinforcement Learning (RL) deals with learning a mapping (a policy) between states and actions by interacting with an environment, in Imitation Learning (IL) a policy is learned from demonstrations. Methods based on IL, also known as Learning from Demonstration (LfD), have shown promise in the field of robotics (Bakker & Kuniyoshi, 1996; Atkeson & Schaal, 1997; Schaal, 1999; Argall et al., 2009; Nair et al., 2018) and games (Silver et al., 2016; Justesen & Risi, 2017; Gudmundsson et al., 2018; Thurau et al., 2004; Gorman & Humphrys, 2007; Vinyals et al., 2019; Harmer et al., 2018).

IL is a form of supervised learning, in which the goal is to learn a policy $\pi(s)$, mapping a state $s \in S$ to a probability distribution over possible actions. In contrast to an RL task, the agent cannot interact with the environment during training but is instead presented with a dataset $D$ of demonstrations. A demonstration $d_j \in D$ consists of $k_j$ sequential state-action pairs, where the action was taken in the state by some policy. While not a general requirement, in this paper a demonstration corresponds to an episode, i.e. starting from an initial state and ending in a terminal state.

Generative Adversarial Imitation Learning (GAIL) uses a Generative Adversarial Network (GAN) architecture wherein the generator is a policy that produces trajectories (without access to rewards) and the discriminator has to distinguish between the generated trajectories and trajectories from a

set of demonstrations (Ho & Ermon, 2016). Two extensions of GAIL learn a latent space of the demonstrations (Wang et al., 2017; Li et al., 2017), which results in a conditioned policy similarly to our approach. While our approach requires manually designed behavioral dimensions, this can give the user more control over the learned policy; different behavioral spaces can be beneficial for different purposes. A latent space also does not explicitly bare meaning, in contrast to a manually defined behavioral space. Additionally, our approach learns a low-dimensional behavioral space that is suitable for fast adaptation.

## 2.2 Quality Diversity & Behavioral Repertoires

Traditional optimization algorithms aim at finding the optimal solution to a problem. Quality Diversity (QD) algorithms, on the other hand, attempt to find a set of high-performing solutions that each behave as differently as possible (Pugh et al., 2016). QD-algorithms usually rely on evolutionary algorithms, such as Novelty Search with Local Competition (NSLC) (Lehman & Stanley, 2011b) or MAP-Elites (Mouret & Clune, 2015). NSLC is a population-based multi-objective evolutionary algorithm with a novelty objective that encourages diversity and a local competition objective that measures an individual's ability to outperform similar individuals in the population. Individuals are added to an archive throughout the optimization process if they are significantly more novel than previously explored behaviors. MAP-Elites does not maintain a population throughout the evolutionary run, only an archive divided into cells that reflect the concept of behavioral niches in a pre-defined behavioral space. For example, in Cully & Mouret (2016) different cells in the map correspond to different walking gaits for a hexapod robot. Both NSLC and MAP-Elites results in an archive of diverse and high-performing solutions. The pressure toward diversity in QD-algorithms can help the optimization process escape local optima (Lehman & Stanley, 2011a), while the diverse set of solutions also allows for online adaption by switching intelligently between these (Cully et al., 2015). We will describe variations of such an adaption procedure in the next section.

QD is related to the general idea of learning behavioral repertoires. Where QD-algorithms optimize towards a single quality objective by simultaneously searching for diversity, a behavioral repertoire can consist of solutions optimized towards different objectives as in the Transferability-based Behavioral Repertoire Evolution algorithm (TBR-Evolution) by Cully & Mouret (2016).

## 2.3 Bandit Algorithms & Bayesian Optimization

Given either a discrete set or a continuous distribution of options, we can intelligently decide which options to select to maximize the expected total return over several trials. To do this, we consider the discrete case as a $k$-armed bandit problem and the continuous case as a Bayesian optimization problem. In the continuous case, the problem can also be simplified to a $k$-armed bandit problem, simply by picking $k$ options from the continuous space of options.

The goal of a $k$-armed bandit problem is to maximize the total expected return after some number of trials by iteratively selecting one of $k$ arms/options, each representing a fixed distribution of returns (Sutton & Barto, 2018). To solve this problem, one must balance exploitation (leveraging an option that has rendered high returns in the past) and exploration (trying options to gain a better estimation of their expected value). A bandit algorithm is a general solution to $k$-armed bandit problems. One of the most popular of these is the Upper Confidence Bound 1 (UCB1) algorithm (Auer et al., 2002) that first tries each arm once and then always selects the option that at each step maximizes:

$$\overline{X}_j + C\sqrt{\frac{2\ln t}{n_j}}, \tag{1}$$

where $\overline{X}_j$ is the mean return when selecting option $j$ after $t$ steps, $n_j$ is the number of times option $j$ has been selected, and $C$ is a constant that determines the level of exploration.

This $k$-armed bandit approach can be considered as the discrete case of the more general approach of Bayesian optimization (BO), in which a continuous black-box objective function is optimized. BO starts with a prior distribution over objective functions, which is then updated based on queries to the black-box function using Bayes' theorem. The Intelligent Trial and Error algorithm (IT&E) uses BO for robot adaptation to deal with changes in the environment by intelligently searching in the continuous behavioral space of policies found by MAP-Elites (Cully et al., 2015). In their approach,

the fitness of all solutions in the behavioral space is used to construct a prior distribution of the fitness, which is also called a behavior-performance map. A Bayesian optimizer is then used to sample a point in the map, record the observed performance, and compute a posterior distribution of the fitness. This process is continued until a satisfying solution is found. IT&E could also be applied as an adaptation procedure to the policy found by BRIL, by creating a prior distribution based on some quality information of the demonstrations, e.g. player rating or win-rate. A Bayesian optimizer is then used to optimize the behavioral feature input. In this paper we will, however, simplify the adaption process to a discrete $k$-armed bandit problem with $k$ manually selected behavioral features and leave the use of Bayesian optimization techniques for future work.

## 2.4 DIMENSIONALITY REDUCTION

In this paper, the dimensionality of the behavior space is reduced using the t-distributed Stochastic Neighbor Embedding (t-SNE) by Maaten & Hinton (2008). In Stochastic Neighbor Embedding (SNE, a precursor to t-SNE), a graph is embedded by minimizing the distance between two probability distributions measured with the Kullback-Leibler divergence. The first of these probability distributions reflect the similarity between the high-dimensional points, and the second one measures the similarity between the embedded, low dimensional points. The high-dimensional probability is fixed, while the embedding is iteratively updated to minimize the distance between its probability distribution and the fixed one. t-SNE uses a Student t-distribution kernel for the embedding's probability, solving previously known flukes of SNE such as the crowding problem.

Even though t-SNE is considered state-of-the-art in dimension reduction, several other techniques could be explored. For instance, certain datasets' structure can be recovered in low dimensional space using simpler algorithms such as Principal Component Analysis (PCA). Otherwise, other methods such as Isometric Feature Mapping (Isomap) by Tenenbaum et al. (2000), Locally Linear Embedding (LLE) by Saul & Roweis (2001) and Uniform Manifold Approximation and Projection (UMAP) by McInnes et al. (2018) could be used.

## 2.5 UNIVERSAL POLICIES

In value-based RL, one typically learns a state value function $V_\pi(s)$ or a state-action value function $Q_\pi(s, a)$ for a policy $\pi$. Universal Value Function Approximators (UVFA) instead learn a joint distribution $V_\pi(s, g)$ or $Q_\pi(s, a, g)$ over all goals $G$ (Schaul et al., 2015). UVFA can be learned using supervised learning from a training set of optimal values such as $V_g^*(s)$ or $Q_g^*(s, a)$, or it can be learned through RL by switching between goals both when generating trajectories and when computing gradients. Hindsight Experience Replay is an extension to UVFAs, which performs an additional gradient update with the goal being replaced by the terminal state; this modification can give further improvements when it is infeasible to reach the goals (Andrychowicz et al., 2017). An extension to Generative Adversarial Imitation Learning (GAIL) augments each trajectory with a *context* (Merel et al., 2017), which specifies the agent's sub-goals that can be modulated at test-time.

In our approach, we are not considering goals, but rather behaviors, intending to learn a universal policy $\pi(s, b)$ over states $s \in S$ and behaviors $b \in B$ in a particular behavioral space. We are thus combining the QD approach of designing a behavioral space with the idea of learning a universal policy to express behaviors in this space.

## 3 BEHAVIORAL REPERTOIRE IMITATION LEARNING (BRIL)

This section describes two approaches to learning behavioral repertoires using IL. We first describe how a behavioral space can be formed from demonstrations. Then we introduce a naive IL approach that first clusters demonstrations based on their coordinates in the behavioral space, and then applies traditional IL on each cluster. Finally, BRIL is introduced, which learns a single policy augmented with a behavioral feature input rather than learning multiple policies for each behavioral cluster.

### 3.1 BEHAVIORAL SPACES FROM DEMONSTRATIONS

A behavioral space consists of some behavioral dimensions that are typically determined by the experimenter. For example, in StarCraft, behavioral dimensions can correspond to the ratio of each

army unit produced throughout the game to express the strategic characteristics of the player. A behavioral space can require numerous dimensions to be able to express meaningful behavioral relationships between interesting solutions for a problem. Intuitively, if the problem is complex, more dimensions can give a finer granularity in the diversity of solutions. However, there is a trade-off between granularity and adaptation, as low-dimensional spaces are easier to search in. We thus propose the idea of first designing a high-dimensional behavioral space and then reducing the number of dimensions through dimensionality reduction techniques. In our preliminary experiments, it has shown beneficial to reduce the space to two dimensions, as it allows for easy visualization of the data distribution and it also seems to be a good trade-off between granularity and adaptation speed. In preliminary experiments with one-dimensional behavioral spaces, we noticed that nearby solutions could be wildly different.

### 3.2 Imitation Learning on Behavioral Clusters

The naive IL approach for learning behavioral repertories trains $n$ policies on $n$ behaviorally diverse subsets of the demonstrations. This idea is similar to the state-space clustering in Thurau et al. (2004), but here we cluster data points in a behavioral space instead. When a behavioral space is defined, each demonstration can be specified by a particular behavioral description (a coordinate in the $\mathbb{R}^n$ dimensional space), where afterward a clustering algorithm can split the dataset into several subsets. Hereafter, traditional IL can be applied to each subset to learn one policy for each behavioral cluster. This approach creates a discrete set of policies similar to current QD algorithms. However, it introduces a dilemma: if the clusters are small, there is a risk of overfitting to these reduced training sets. On the other hand, if the clusters are large but few, the granularity of behaviors is lost.

### 3.3 Learning Behavioral Repertoires

QD algorithms typically fill an archive with diverse and high-quality solutions, sometimes resulting in thousands of policies stored in a single run, which increases the storage requirements in training as well as in deployment. To reduce the storage requirement, one can decrease the size of the archive, with the trade-off of losing granularity in the behavioral space. The main approach introduced in this paper, called Behavioral Repertoire Imitation Learning (BRIL), solves these issues and reduces overfitting by employing a universal policy instead, in which a single policy is conditioned on a behavioral description. In contrast to QD algorithms, the goal of BRIL is neither to optimize quality nor diversity directly. Instead, BRIL attempts to imitate and express the diverse range of behaviors and the quality that exists in a given set of demonstrations. Additionally, BRIL produces a continuous space of policies which is potentially more expressive than a discrete set.

BRIL extends the traditional imitation learning setting through the following approach. First, the behavioral characteristics of each demonstration are determined. If the dimensionality of these descriptions is large, it can be useful to reduce the space as described in the earlier section. A training set of state-action-behavior triplets is then constructed, such that the behavior is equal to the behavioral description of the corresponding demonstration. Then, a policy $\pi(s, b)$ is trained in a supervised way on this dataset to map states and behaviors to actions. Following this approach, the training set is not reduced to small behavioral clusters.

When the trained policy is deployed, the behavioral feature input can be modulated to manipulate its behavior. The simplest approach is to fix the behavioral features throughout an episode, evaluate the episodic return, and then consider new behavioral features for the next episode. This approach should allow for episodic, or inter-game, adaptivity, which will be explored in our experiments. One could also manipulate the behavioral features during an episode e.g. by learning a meta-policy.

## 4 Experiments

This section presents the experimental results of applying BRIL to the game of StarCraft. Policies are trained to control the build-order planning module of a relatively simple scripted StarCraft bot[1] that plays the Terran race. While the policy is trained off-line, our experiments attempt to optimize the playing strength of this bot online, in-between episodes/games, by manipulating its behavior.

---

[1][REDACTED for anonymity]

### 4.1 BEHAVIORAL FEATURE SPACE

The behavioral space for a StarCraft build-order policy can be designed in many ways. Inspired by the AlphaStar League Strategy Map (Vinyals et al., 2019), the behavioral features are constructed from the army composition, such that the dimensions represent the ratios of each unit type. We achieve this by traversing all demonstrations in the data set, counting all the army unit creation events, and computing the relative ratios. Each demonstration thus has an $n$-dimensional behavioral feature description, where $n = 15$ is the number of army unit types for Terran.

To form a 2D behavioral space, which allows for easier online search and analysis, we apply t-Distributed Stochastic Neighbor Embedding (t-SNE). Fig. 2 visualizes the points of all the demonstrations in this 2D space and Fig. 2a shows four plots where the points are colored to show the ratios of Marines, Marauders, Hellions, and Siege Tanks that were produced during these games.

### 4.2 CLUSTERING

For the baseline approach that applies IL to behavioral clusters, we use density-based spatial clustering of applications with noise (DBSCAN) with $\epsilon = 0.02$ and a minimum number of samples per cluster of 30. We performed a grid-search on these two parameters to find the most meaningful data separation; however, the clustering is not perfect due to the many outliers. The clusters are visualized in Fig. 2b, with outliers shown in black.

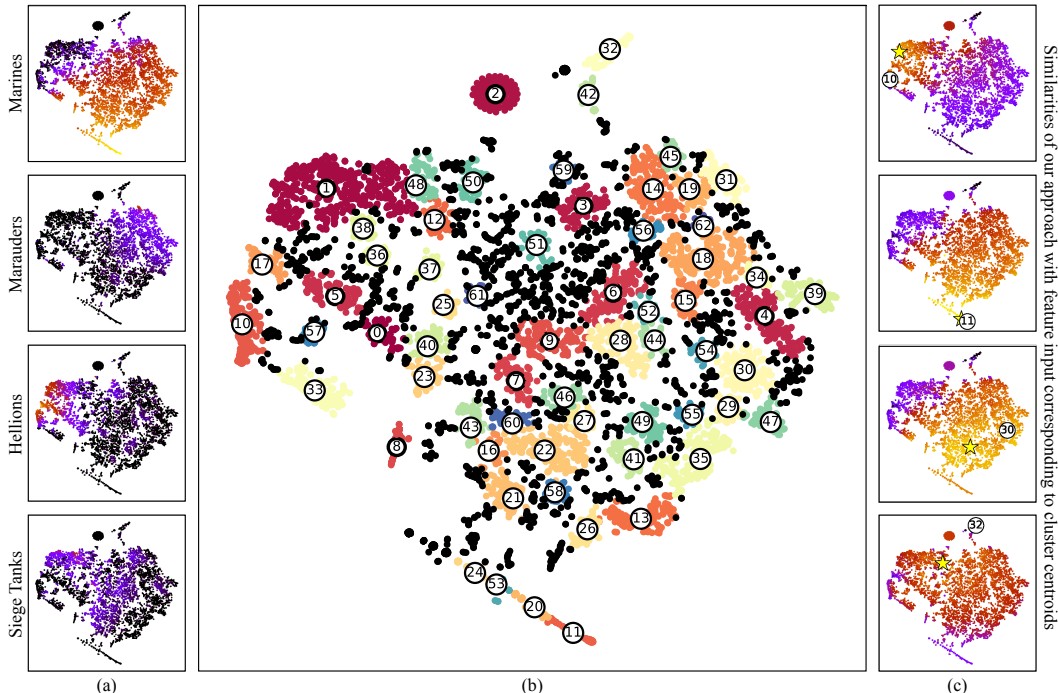

Figure 2: **Visualizations of the 2D behavioral space of Terran army unit combinations in 7,777 Terran versus Zerg replays.** Each point represents a replay from the Terran player's perspective. The space was reduced using t-SNE. (a) The data points are illuminated (black is low and yellow is high) by the ratio of Marines, Marauders, Hellions, or Siege Tanks produced in each game. (b) 62 clusters found by DBSCAN. Cluster centroids are marked with a circle and the cluster number and outliers are black. The noticeable cluster 2 has no army units. (c) The similarity between the behaviors of the human players and our approach with four different feature inputs, corresponding to the coordinates of centroids of cluster 10, 11, 30, and 32. The behavior of our approach is averaged over 100 games against the easy Zerg bot and its nearest human behavior is marked with a star. The behavior of the learned policy can be efficiently manipulated to change its behavior. Additionally, we can control the behavior such that it resembles the behavior of a human demonstration.

### 4.3 PERFORMANCE IN STARCRAFT

We trained three groups of neural networks, all with three hidden layers and 256 hidden nodes per layer: (1) One baseline model trained on the whole dataset with no augmentation of behavioral features, (2) a BRIL model on the whole dataset with two extra input nodes for the behavioral features (i.e. the coordinates in Fig. 2b), and (3) several cluster baseline models trained only on demonstrations from their respective clusters without the augmented behavioral features.

We applied these trained policy models as build-order modules in the scripted StarCraft II Terran bot sc2bot. It is important to note that this is a very simplistic bot with several flaws and limitations. Therefore the main goal in this paper is not to achieve human-level performance in StarCraft, but rather to test if BRIL allows us to do manipulate its behavior and enables online adaptation. The build-order module, here controlled by one of our policies, is queried with a state description and returns a build-order action, i.e. which building, research, or unit to produce next. The worker and building modules of the bot perform these actions accordingly, while assault, scout, and army modules control the army units. Importantly, policies we test act in a system that consists of both the bot, the opponent bot, and the game world. When we want to utilize our method for adaptation, we are thus not only adapting to the opponent but also the peculiarities of the bot itself.

| Method | Wins | Distance to cluster centroid | | | | Combat units produced | | | | |
|---|---|---|---|---|---|---|---|---|---|---|
| | | C10 | C11 | C30 | C32 | Marines | Marauders | Hellions | S. Tanks | Reapers |
| IL | 41/100 | 0.58 | 0.22 | 0.39 | 0.75 | $44.1 \pm 50.5$ | $0.7 \pm 3.2$ | $2.6 \pm 7.6$ | $1.7 \pm 6.5$ | $0.3 \pm 1.1$ |
| IL (C10) | 3/100 | **0.05** | 0.76 | 0.81 | 0.71 | $1.1 \pm 2.3$ | $0.1 \pm 0.3$ | **3.11 ± 6.1** | **0.1 ± 0.4** | $0.1 \pm 0.33$ |
| IL (C11) | 7/100 | 0.74 | **0.00** | 0.52 | 0.96 | $18.8 \pm 38.4$ | $0.0 \pm 0.0$ | $0.0 \pm 0.0$ | $0.0 \pm 0.0$ | $0.0 \pm 0.0$ |
| IL (C30) | 18/100 | 0.76 | 0.21 | **0.31** | 0.79 | **43.5 ± 62.6** | **0.9 ± 5.4** | $0.2 \pm 1.3$ | $0.0 \pm 0.2$ | $0.2 \pm 0.8$ |
| IL (C32) | 0/100 | 0.71 | 0.94 | 0.57 | **0.04** | $0.1 \pm 0.2$ | $0.0 \pm 0.0$ | $0.0 \pm 0.0$ | $0.0 \pm 0.0$ | **9.9 ± 18.5** |
| BRIL (C10) | 27/100 | **0.21** | 0.85 | 0.81 | 0.60 | $2.4 \pm 4.9$ | $0.0 \pm 0.0$ | **14.6 ± 18.9** | $4.0 \pm 5.2$ | $0.2 \pm 0.6$ |
| BRIL (C11) | **76/100** | 0.70 | **0.05** | 0.53 | 0.95 | **81.4 ± 50.1** | $0.0 \pm 0.1$ | $0.2 \pm 0.1$ | $0.9 \pm 2.4$ | $0.3 \pm 0.6$ |
| BRIL (C30) | 47/100 | 0.60 | 0.31 | **0.29** | 0.65 | $41.6 \pm 36.4$ | **2.4 ± 6.7** | $0.7 \pm 2.5$ | $4.1 \pm 7.8$ | $0.5 \pm 1.2$ |
| BRIL (C32) | 16/100 | 0.42 | 0.72 | 0.53 | **0.36** | $7.1 \pm 11.4$ | $1.7 \pm 6.5$ | $3.2 \pm 8.3$ | **6.7 ± 9.7** | **0.8 ± 1.5** |
| Method | Win | Wins for each option | | | | Combat units produced | | | | |
| | | C10 | C11 | C30 | C32 | Marines | Marauders | Hellions | Siege Tanks | Reapers |
| BRIL (UCB1) | 61/100 | 5/14 | **47/59** | 8/18 | 1/9 | $52.1 \pm 47.7$ | $0.5 \pm 3.2$ | $4.3 \pm 12.5$ | $2.5 \pm 6.2$ | $0.3 \pm 1.3$ |

Table 1: Results in StarCraft using Imitation Learning (IL) on the whole training set, IL on individual clusters (C10, C11, C30, and C32), Behavioral Repertoire Imitation Learning (BRIL) with fixed behavioral features corresponding to centroids in C10, C11, C30, and C32. Additionally, results are shown in which UCB1 selects between the four behavioral features in-between games. Each variant played 100 games against the easy Zerg bot. The nearest demonstration in the entire dataset was found based on the bot's mean behavior (normalized army unit combination) and the distance to each cluster centroid are shown. The results demonstrate that by using certain behavioral features, the BRIL policy outperforms the traditional IL approach as well as IL on behavioral clusters.

We will first focus on the results of the traditional IL approach. Table 1 shows the number of wins in 100 games on the two-player map CatalystLE as well as the corresponding average behaviors (i.e. the army unit ratios). Our bot played as Terran against the built-in Easy Zerg bot. The traditional IL approach won 41/100 games. IL on behavioral clusters showed very poor performance with a maximum of 18/100 wins by the model trained on C30. Besides the number of wins, we compute the nearest demonstration in the entire data set from the average behavior and use it as an estimate of the policy's position in the 2D behavioral space. From the estimated point, we calculate the distance to each of the four cluster centroids. This analysis revealed that the policies trained on behavioral clusters express behaviors close to the clusters they were trained on (see the distances to the cluster centroids in Table 1). We hypothesize that the poor win rates of this naive approach are due to their training sets being too small such that the policies do not generalize to many of the states explored in the test environment.

Table 1 also shows the results for BRIL with the coordinates of the four cluster centroids as behavioral features. BRIL (C11) achieves a win rate of 76/100, thus outperforming traditional IL. These results demonstrate that, for some particular environment, the model can be tuned to achieve a higher performance than traditional IL. Analyzing the behavior of the bot with the behavioral features of C11 reveals that it performs an all-in Marine push, similarly to the behavior of the demonstrations

in C11 (notice the position of C11 on Fig. 2.b and the illumination of Marines on Fig. 2.a). With the behavioral features of C30, the approach reached a higher win rate than traditional IL; however, this difference was not significant. We also notice that for both BRIL and IL on behavioral clusters, the average expressed behavior is closest to the cluster centroid that it was modulated to behave as, among the four clusters we selected. The results show that the behavior of the learned BRIL policy can be successfully controlled. However, the distances are on average larger than for IL on behavioral clusters.

### 4.4 ONLINE ADAPTATION

The final test aims to verify that we can indeed use BRIL for online adaptation. We apply the UCB1 algorithm to select behavioral features from the discrete set of four options: {C10, C11, C30, C32} (i.e. the two-dimensional feature descriptions of these cluster centroids). This approach enables the algorithm to switch between behavioral features in-between games based on the return of the previous one, which is 1 for a win and 0 otherwise. The adaptive approach achieves 61/100 wins by identifying the behavior of C11 as the best option. Not surprisingly, the win rate is lower than when having the behavioral features of C11 fixed, while it outperforms traditional IL.

## 5 DISCUSSION

We proposed two new IL methods in this paper, one which learns a policy that is trained on only one behavioral cluster of data points and one which learns a single modifiable policy on the whole dataset. Our results suggest that policies trained on small behavioral clusters overfit and are thus unable to generalize beyond the states available in the cluster. This drawback might be solved with fewer and larger clusters at the cost of losing granularity in the repertoire of policies. If data is abundant, this approach may also work better while we still suspect the same overfitting would occur. BRIL, on the other hand, is simple to implement and results in a continuous distribution of policies by adjusting the behavioral features. Additionally, the results suggest that BRIL generalizes better, most likely because it learns from the whole training set. However, that generality potentially comes with the cost of higher divergence between the expected behavior (corresponding to the behavioral input features) and the resulting behavior when tested. While an important concern, a divergence is somewhat expected since the test environment is very different from that of the training set (different maps and opponents).

Previous work showed how IL can kick-start learning before applying RL (Silver et al., 2016; Vinyals et al., 2019). With BRIL, one can easily form a population of diverse solutions instead of just one, which may be a promising approach for domains with a plethora of strategic choices like StarCraft. Promising future work could thus combine BRIL with ideas from AlphaStar to automatically form the initial population of policies used in the AlphaStar League.

## 6 CONCLUSIONS

We introduced a new method called Behavioral Repertoire Imitation Learning (BRIL). By labeling each demonstration $d \in D$ with a behavior descriptor confined within a pre-defined behavioral space, BRIL can learn a policy $\pi(s, b)$ over states $s \in S$ and behaviors $b \in B$. In our experiments, a low-dimensional representation of the behavioral space was obtained through dimensionality reduction. The results in this paper demonstrate that BRIL can learn a policy that, when deployed, can be manipulated by conditioning it with a behavioral feature input $b$, to express a wide variety of behaviors. Additionally, the observed behavior of the policy resembles the behavior characterized by $b$. Furthermore, a BRIL trained policy can be optimized online by searching for optimal behavioral features in a given setting. In our experiments, a policy trained with BRIL was optimized online beyond the performance reached by traditional IL, using UCB1 to select among a set of discrete behavioral features.

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

## A APPENDIX

### A.1 PREDICTION ACCURACY

Data from StarCraft 2 replays are extracted with sc2reaper[2], a tool built using the StarCraft II Learning Environment inspired by the MSC Database (Wu et al., 2017). 7,777 replays of Terran vs. Zerg were processed, extracting state-action pairs every half a second resulting in a dataset of 1,625,671 state-action pairs.

These states contain an abstraction of the game state similar to the one found in Justesen & Risi (2017). This abstraction includes: (1) the agent's resources, supply, units and technologies, (2) a tally of the enemy's units that have been observed, and (3) the agent's units and technologies in production, including how far they are from being completed.

---

[2][REDACTED for anonymity]

| Method | Mean test accuracy | Mean test loss | # of replays |
|--------|-------------------|----------------|--------------|
| IL | $48.090 \pm 0.080$ | $1.775 \pm 0.003$ | 7777 |
| BRIL | $\mathbf{48.167 \pm 0.083}$ | $\mathbf{1.768 \pm 0.003}$ | 7777 |
| IL (C10) | $32.608 \pm 0.417$ | $2.096 \pm 0.013$ | 189 |
| IL (C11) | $71.326 \pm 0.316$ | $1.088 \pm 0.008$ | 74 |
| IL (C30) | $45.709 \pm 0.499$ | $1.976 \pm 0.020$ | 149 |
| IL (C32) | $45.855 \pm 0.681$ | $1.770 \pm 0.006$ | 97 |

Table 2: Test accuracy and loss for IL, BRIL, and IL trained on clusters 10, 11, 30 and 32. Results show no significant difference between the IL and BRIL in terms of prediction accuracy. BRIL is, however, able to express multiple behaviors based on the additional input (see Table 1).

Once the replays were post-processed for clustering, the dataset was split into training/test/validation following a 60% / 30% / 10% split per cluster. With this splitting, the three groups of models that were discussed in subsection 4.3 were trained.

These experiments were carried out ten times per model. Table 2 shows the mean test accuracy and mean test loss over these ten models for the baseline (IL) approach, the novel BRIL approach, and four different cluster baselines (Clusters 10, 11, 30 and 32), which were selected for their wildly different behaviors. The results show that augmenting by behavioral features has no significant effect on the test accuracy or loss. However, the next section shows how our new approach is able to express different behaviors with a single neural network.

## A.2 STARCRAFT

Video games are popular testbeds for the development and testing of AI algorithms (Justesen et al., 2019). Real-Time Strategy (RTS) games, such as StarCraft, are among the hardest games for algorithms to learn as they contain a myriad of problems, such as dealing with imperfect information, adversarial real-time planning, sparse rewards, or huge state and action spaces (Buro, 2003). Several algorithms and bots have been built by the AI community (Ontanón et al., 2013; Churchill et al., 2016) to compete in tournaments such as the AIIDE StarCraft AI Competition[3], the CIG StarCraft RTS AI Competition[4] and the Student StarCraft AI Competition[5].

This paper deals with the problem of learning a policy for build-order planning in StarCraft. Similarly to the work by Justesen & Risi (2017), a neural network-based policy is trained using IL from state-action pairs, and the policy is then combined with a bot with scripted procedures for low-level tasks. However, the network in the work by Justesen & Risi (2017) learns an "average" policy, while the approach introduced in this paper is able to learn a behavioral repertoire from demonstrations. The build-order planning problem has also been approached with RL, optimizing a build-order policy for a specific bot (Tang et al., 2018; Sun et al., 2018).

The StarCraft II Learning Environment (SC2LE) is an API that allows scripted bots and RL algorithms to interact with the game (Vinyals et al., 2017). SC2LE also allows data extraction thereby enabling IL from human demonstrations (Wu et al., 2017). AlphaStar is a recent approach that learns a human-level AI policy for StarCraft II using a combination of IL and RL (Vinyals et al., 2019). First, a neural network was trained from supervised demonstrations. The IL model was then used to seed a tournament with several RL agents (called the AlphaStar League), in which the agents competed. This approach resulted in a series of competitive and diverse behaviors. Despite the skill level of AlphaStar, it does not offer a solution for inter-game adaptation and the diversity of the AlphaStar League has to be managed carefully.

## A.3 ARMY COMPOSITIONS IN-GAME

Replays in each of the clusters we examined in the bandit problem (that is, C10, C11, C30 and C32) exhibit a particular army composition in the game. Cluster 11, for example, shows a strategy com-

---

[3]http://www.cs.mun.ca/~dchurchill/starcraftaicomp/
[4]http://cilab.sejong.ac.kr/sc_competition/
[5]http://sscaitournament.com/

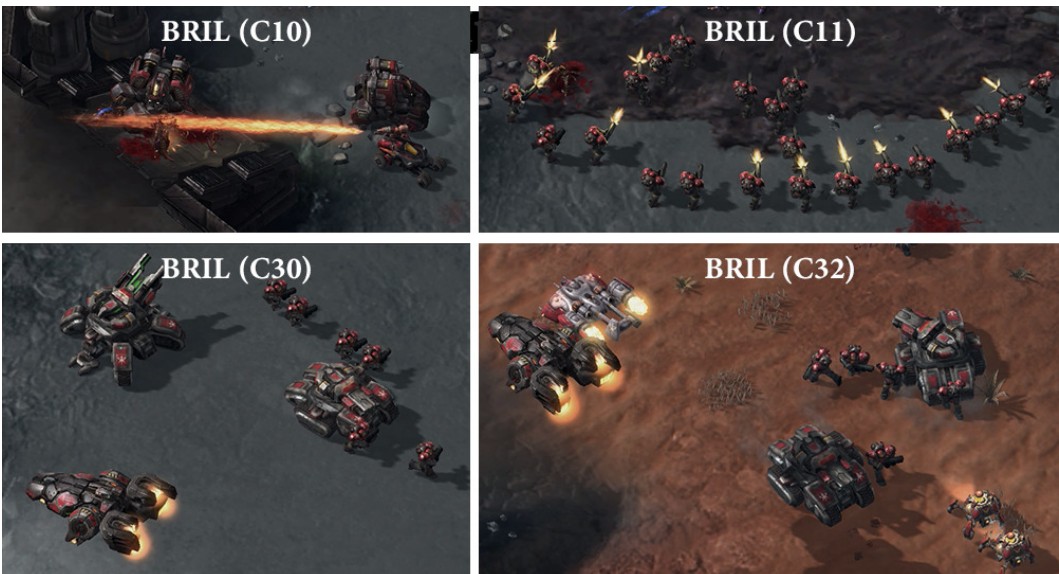

Figure 3: Screenshots of typical army compositions produced by our trained BRIL policy with behavioral features corresponding to the centroids of cluster 10, 11, 30 and 32. BRIL (C10) executes early timing pushes with Hellions and Cyclones, BRIL (C11) is aggressive with Marines only, BRIL (C30) creates mixed armies with many Marines and Siege Tanks, and BRIL (C32) also creates mixed armies but with less Marines and more Widow Mines.

posed almost purely of Marines (see Fig. 2). Fig. 3 shows screenshots of typical army compositions produced by the BRIL policy with the four different behavioral features.

