# OpenReview forum: "Learning a Behavioral Repertoire from Demonstrations"
_ICLR.cc/2020/Conference — Reject_

### Official Review · AnonReviewer2 · 2019-10-18
**Official Blind Review #2**

**Rating:** 1

**Review:**

Summary:
The paper proposes behavioral repertoire imitation learning (BRIL) which aims to learn a collection of policy from diverse demonstrations. BRIL learns such a collection by learning a context-dependent policy, where the context variable represents behavior of each demonstration. To obtain a context variable, BRIL rely on user’s knowledge, where the user manually defines a feature space that describes behavior. This feature space is then reduced by using a dimensionality reduction method such as t-SNE. Lastly, the policy is learned by supervised learning (behavior cloning) with a state-context input variable and an action output variable. The method is experimentally evaluated on the StarCraft environment. The results show that BRIL performs better than two baselines: behavior cloning on diverse demonstrations and behavior cloning on clustered demonstrations.

Score:
The weaknesses of the paper are novelty, clarity, and evaluation. Please see the detailed comments below. I vote for rejection.

Comments:
- Novelty of the proposed idea.
The major issue of the paper is the lack of novelty. The idea of learning a context-dependent policy in BRIL closely resembles that of existing multi-modal IL methods (Wang et al., 2017, Li et al., 2017). The main difference is that, BRIL relies on a manually defined context variable (behavioral feature space). In contrast, the existing methods aim to learn the context variable from demonstrations. BRIL is too simple when compared to the existing methods. Moreover, using a manually specified feature space does not go well with the main principle of deep learning, which is to learn informative feature spaces from data end-to-end. I think that ICLR is not a suitable venue for this paper.

- Clarity of the proposed method.
The second issue of the paper is clarity. Specifically, two important steps of BRIL is policy learning by supervised learning (behavior cloning) and dimensionality reduction by t-SNE. However, explanations of these two steps are vague and incomplete. For example, in Section 2.1, the paper describes IL as supervised learning, but does not mention the issue of covariate shift, which is well-known when treating IL as supervised learning (Ross et al., 2011). Also, it is incorrect to state that an IL agent cannot interact with the environment during training, since many IL methods such as GAIL require interactions with the environment during training. Meanwhile, in Section 2.4, it is unclear how probability distributions in t-SNE reflect similarity between data points.

[1] Stéphane Ross, Geoffrey Gordon, and Drew Bagnell. A reduction of imitation learning and structured prediction to no-regret online learning. AISTATS, 2011.

- Evaluation of the proposed method is too narrow.
The paper lacks important baseline methods in the experiment. Specifically, the paper does not compare BRIL against multi-modal IL methods (Wang et al., 2017, Li et al., 2017) which also learn a context-dependent policy. Moreover, BRIL is evaluated only on the StarCraft environment with only one kind of manually specified feature. This raises a question of generality and sensitivity against the choice of feature of BRIL. To improve the paper, I suggest the authors to compared the proposed method against multi-modal IL methods on different environment, and evaluate BRIL with different choices of the behavioral features.



**Experience Assessment:**

I have published one or two papers in this area.

**Review Assessment: Checking Correctness Of Derivations And Theory:**

N/A

**Review Assessment: Checking Correctness Of Experiments:**

I assessed the sensibility of the experiments.

**Review Assessment: Thoroughness In Paper Reading:**

I read the paper at least twice and used my best judgement in assessing the paper.

---

### Official Review · AnonReviewer1 · 2019-10-23
**Official Blind Review #1**

**Rating:** 3

**Review:**

This work examines the problem of using training a policy which can emulate a variety of different strategies based on a set of demonstrations representing this space of strategies.  The proposed method, BRIL, computes a feature vector for each demonstration, and then employs a dimensionality reduction technique to map the demonstrations to a latent space of strategies.  BRIL then preforms behavioral cloning on these demonstrations, with the reduced representation of the current strategy as an additional input to the policy model.  Empirical evaluation of BRIL is conducted in StarCraft II, where the agent is tasked with scheduling the construction of different units (other aspects of play are controlled by built-in AI).  Results show that when conditioned on good strategies, the BRIL model is superior to a base imitation learning model trained without strategy information.

The best way to view this work is as a method for learning goal conditioned polices from demonstrations.  While the raw data does not distinguish between different strategies, a postprocessing phase generates task description vectors with specific semantics (the ratios of different unit types built) which are given as input to the BRIL model in addition to the current state.  Therefore, rather than identifying the latent space of strategies present in the demonstration data, BIRL learns a policy which is parameterized by an externally defined space of target behaviors.  The dimensionality reduction step in the strategy space does recover some latent structure, but this is still with respect to a task-specific space of strategy annotations, rather than the demonstrated behaviors themselves.

While this is not an issue with BRIL itself, it should be made clear in the paper that BRIL is learning policies conditioned on explicit strategy descriptions, so that the work can be properly positioned in the literature.  Neither the theoretical discussion nor the empirical results compare BRIL against existing work on learning goal-conditioned policies.

The empirical results comparing BRIL against a base IL agent are interesting, however, in that BRIL, when conditioned on the best strategy (building mostly marines), actually outperforms IL trained solely on episodes which followed this strategy.  This suggest the possibility that BRIL was able to transfer knowledge from episodes demonstrating other strategies to make up for the limited information available in the demonstrations of the target behavior.

In section 4.4 the paper briefly discusses the idea of evaluating different strategies executed by the BRIL model, and selecting those that perform best, as a means of learning strong policies for the StarCraft task.  The use of human demonstrations to improve the performance of learning algorithms on difficult control tasks is a widely studied problem, and optimization.  This idea is not explored in any great detail however, and no comparisons with existing methods combining IL with RL are conducted, so the value of BRIL in that context is unclear.  I would, however, recommend this as direction for future work with BRIL.

**Experience Assessment:**

I have read many papers in this area.

**Review Assessment: Checking Correctness Of Derivations And Theory:**

I assessed the sensibility of the derivations and theory.

**Review Assessment: Checking Correctness Of Experiments:**

I carefully checked the experiments.

**Review Assessment: Thoroughness In Paper Reading:**

I read the paper at least twice and used my best judgement in assessing the paper.

---

### Official Review · AnonReviewer3 · 2019-10-26
**Official Blind Review #3**

**Rating:** 3

**Review:**

This paper presents Behavioral Repertoire Imitation Learning (BRIL) which is a way to learn a policy via imitation learning that can be modulated with different behavior inputs that adjust the policy's behavior.  Demonstrations used in training are labeled with differences in behavior across dimensions (which are then reduced to two dimensions using t-SNE), and then these behavior labels are provided as additional input when training a NN from demonstrations using behavior cloning.  Experimental results are shown for learning a BRIL policy from 7000+ demonstrations of humans playing StarCraft, and are compared to that of learning a single behavior cloned policy trained on all demonstrations as well as behavior cloned policies trained on subsets of demonstrations clustered by their behavior.

One of the authors' claims is that BRIL is able to learn a policy that can have a wide variety of behaviors based on the behavior input given as input to the policy which is backed by their results.  They also claim that the BRIL model can be tuned to have higher performance than a policy learned by traditional IL which is shown by their results, although I think that may be somewhat of a function of the environment and set of demonstrations.

Another claim by the authors in section 4.3 is that behavior can be successfully controlled, and they say "for both BRIL and IL on behavioral clusters, the average expressed behavior is closest to the cluster centroid that it was modulated to behave as, among the four clusters we selected."  I'm not sure this statement is true as IL (C30) is closer to C11 than C30 in table 1.

The biggest weakness in this paper -- and barrier to acceptance -- is in the really small sample size of the results where only 4 cluster behaviors out of 62 are evaluated.  I would like to see information about the aggregate performance and relative behaviors of BRIL policies compared to the policies trained on the clustered subset of demonstrations across all clusters (or certainly a lot more than just 4 which is less than 10%!) in order to have a better evaluation of the BRIL approach.  The claims of BRIL doing better than clustered IL and being close to a behavior cluster centroid are not that convincing with so few sampled data points.

In section 4.2 how do you define or quantify "the most meaningful data separation" when doing a grid search on parameters for clustering?

Outside of making things easier to visualize, I'm not sure why it is necessary to reduce the dimensionality of the behavior input for BRIL. Reducing the dimensionality also reduces the set of different policy behaviors that can be expressed.  Additionally, if one might want to adjust a behavior such as having more or less of a certain unit type in StarCraft, the reduced behavior space makes it harder and less intuitive to do so.

Showing that UCB can end up selecting the best BRIL policy (out of 4) and thereby resulting in a better cumulative win record than the traditional IL baseline is kind of trivial and doesn't add much to the paper.  Using a Bayesian optimizer as suggested for future work to select the best behavior inputs for the BRIL policy would be more interesting and give more credence for BRIL being useful in discovering better policies.

**Experience Assessment:**

I have read many papers in this area.

**Review Assessment: Checking Correctness Of Derivations And Theory:**

N/A

**Review Assessment: Checking Correctness Of Experiments:**

I carefully checked the experiments.

**Review Assessment: Thoroughness In Paper Reading:**

I read the paper at least twice and used my best judgement in assessing the paper.

---

### Decision · Program_Chairs · 2019-12-19

**Decision:**

Reject

**Comment:**

This paper proposes a way to lean context-dependent policies from demonstrations, where the context represents behavior labels obtained by annotating demonstrations with differences in behavior across dimensions and the reduced in 2 dimensions. Results are conducted in the domain of StarCraft. The main concerns from the reviewers related to the paper’s novelty (as pointed by R2) and experiments (particularly the lack of comparison with other methods and the evaluation of only 4 out of the 62 behaviour clusters, as pointed by R3). As such, I cannot recommend acceptance, as current results do not provide strong empirical evidence about the superiority of the method against other alternatives.